# Grain neighbour effects on twin transmission in hexagonal close-packed materials

M. Arul Kumar[1], I.J. Beyerlein[2], R.J. McCabe[1] & C.N. Tomé[1]

Materials with a hexagonal close-packed (hcp) crystal structure such as Mg, Ti and Zr are being used in the transportation, aerospace and nuclear industry, respectively. Material strength and formability are critical qualities for shaping these materials into parts and a pervasive deformation mechanism that significantly affects their formability is deformation twinning. The interaction between grain boundaries and twins has an important influence on the deformation behaviour and fracture of hcp metals. Here, statistical analysis of large data sets reveals that whether twins transmit across grain boundaries depends not only on crystallography but also strongly on the anisotropy in crystallographic slip. We show that increases in crystal plastic anisotropy enhance the probability of twin transmission by comparing the relative ease of twin transmission in hcp materials such as Mg, Zr and Ti.

[1] Materials Science and Technology Division, Los Alamos National Laboratory, Los Alamos, New Mexico 87545, USA. [2] Theoretical Division, Los Alamos National Laboratory, Los Alamos, New Mexico 87545, USA. Correspondence and requests for materials should be addressed to M.A.K. (email: marulkr@lanl.gov or marulkr@gmail.com).

Hexagonal close-packed (hcp) alloys such as Mg, Ti and Zr alloys are being used in the transportation (lightweight), aerospace (corrosion resistance and low thermal coefficient) and nuclear (corrosion and radiation resistance) industry, respectively. These alloys undergo deformation twinning when strained[1–3]. Twin embryos form preferably at grain boundaries, where stress concentrations and source defects are predominately located[4,5]. They expand in size within the grain under continued straining, reaching a length scale comparable to that of the grain size. However, under continued straining and under the right conditions, twin lamellae can stimulate the formation of another twin on the other side of a grain boundary (GB), appearing as if the twin has propagated across the boundary. One can envision that this process may continue, triggering twins in their neighbours and creating the so-called twin chains or catalytic twins across the grain structure[4,6,7]. In this work, we refer to two twins that are connected at the grain boundaries as adjoining twin pairs (ATPs) and those that comprises three or more connected twins as twin chains. The former is the focus of this study. Whether twins remain within the confines of the parent grain or transmit across grain boundaries greatly affects the mechanical behaviour[8]. Catalytic twins can increase the likelihood of instabilities, such as void nucleation, cracking and premature failure[9–14].

The transmission of {10–12} tensile twins has been reported at different temperature and loading conditions and in a variety of hcp materials including Mg[7], Mg alloys[4,6,15–17], Zr[18], Ti[19] and Ti alloys[20,21]. Transmission of {11–21} tensile twins was recently studied in rhenium[22]. The outstanding common feature seen in all of these studies is that twin transmission (TT) frequency decreases with increases in GB misorientation. Yet, it is also clear that the material affects the propensity for TT. For instance, an apparent cutoff angle above which TT is rare is 50° for Mg[17], 35° for AZ31 (ref. 4) and 25° for rhenium[22]. To date, only geometric variables have been considered in determining whether twins stop or transmit across grain boundaries[6,15–17,19,20]. How basic material properties affect transmissibility is unknown and establishing direct relationships between TT and material properties can be partly obscured by the fact that twinning occurs heterogeneously across the microstructure[7]. Thus, to examine properly TT, data sets ought to contain at least hundreds to thousands of events.

The aim of this work is to present experimental evidence that material slip characteristics affect TT, and to perform three-dimensional (3D) full-field crystal-plasticity calculations to elucidate the key intrinsic material variables. We obtain statistically significant data sets of transmitted twins in Mg and Zr via an automated electron backscattered diffraction (EBSD) code[7,18]. The combined experimental analysis and calculations reveal that plastic anisotropy (PA) in slip promotes TT: the higher the PA, the more likely TT is for the same misorientation angle and the larger the cutoff misorientation angle. The calculations also forecast an interesting 'boost' effect, whereby for a special set of low-misorientation GBs, TT is more likely than propagating the same twin in its parent grain. These findings reveal another fundamental material property that can be potentially tuned to achieve formable hcp materials.

## Results

**TT in Mg and Zr**. The materials examined are high-purity polycrystalline Mg and Zr with similar initial textures and, hence, similar GB-misorientation distributions[7,18]. The Mg has a strong basal texture resulting from rolling, where most of the basal poles are aligned within 30° of the normal direction of the sheet. Zr also has a strong basal texture, which was processed via clock rolling rather than conventional rolling[23]. Both materials were compressed at $10^{-3}$ per s along an in-plane direction to activate {1012} twinning. To develop a sufficient number of incipient twins in many grains, Zr was deformed at 76 K[18] and Mg was deformed at room temperature[7]. The amount of plastic strain at which the samples were examined was relatively small, 3% for Mg and 5–10% for Zr, at least an order of magnitude below their failure strain. At higher strain levels, twins tend to dominate the microstructure, making it difficult to identify the source GBs.

EBSD was used to map the orientations within these deformed microstructures[24]. Under their respective temperature, strain-rate conditions and strain levels, the materials twinned significantly such that many grains contained fine twins that were visible in EBSD. An automated EBSD technique was employed to locate all GBs that are crossed by ATPs and calculate their misorientation angle (the minimum misorientation accounting for all crystal symmetries)[24]. Large data sets were generated by analysing several distinct EBSD scans[7,18]. The number of grains and twins investigated totalled 2,339 and 8,550 for Mg, and 639 and 1,065 for Zr. Large numbers are critical since only a fraction of all twins are ATPs.

Figure 1 shows a representative scan from each material, exposing some examples of ATPs, where two twins join across the GB. ATPs could have formed in one of two ways: (1) the twins formed simultaneously from a GB or (2) they formed in sequence, where the twin in one grain transmitted to the neighbouring

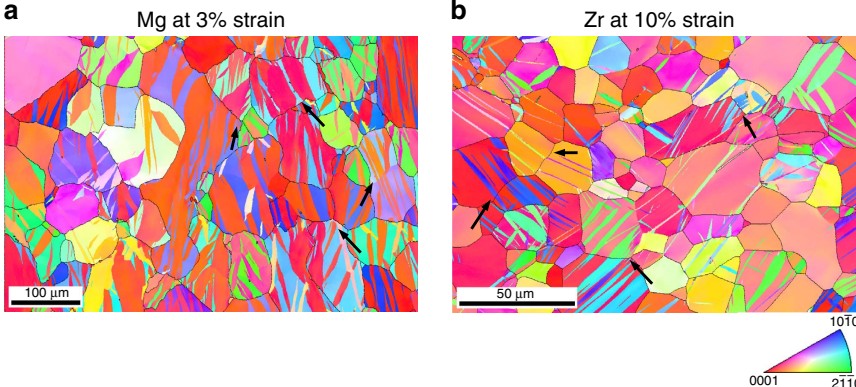

**Figure 1 | Evidence of adjoining twin pairs in the deformed microstructure.** Electron back scattered diffraction (EBSD) scans of (**a**) Mg strained to 3% in-plane compression at room temperature (standard stereographic triangle showing compression direction) and (**b**) Zr strained to 10% in-plane compression at 76 K temperature (standard stereographic triangle showing plate normal direction) at a strain rate of $10^{-3}$ s. Several adjoining twin pairs can be seen; a few are marked by black arrows.

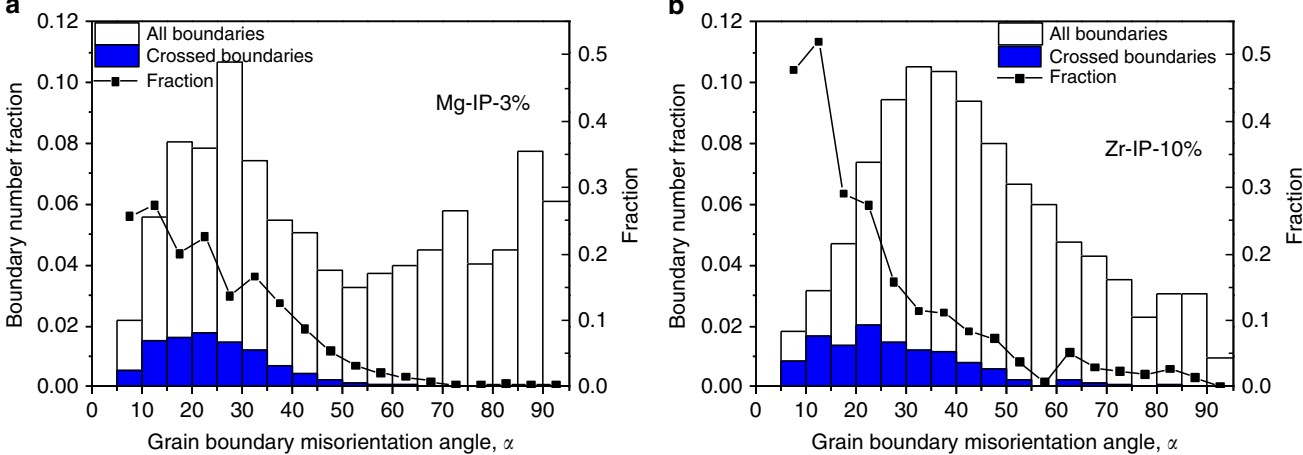

**Figure 2 | EBSD-based evolution of twin transmission frequency.** EBSD-based distribution of grain boundaries and twin crossing at grain boundaries with grain boundary misorientation angle for (**a**) Mg and (**b**) Zr. White bars: all grain boundaries; blue bars: twin-crossed boundaries. The secondary vertical axis shows the fraction of grain boundaries that have been crossed by twins. Non-zero crossed boundary fraction beyond 60° for Zr can be a consequence of local stress deviations associated with clusters of small-sized grains where twin variants with negative 'macroscopic' Schmid factor are activated.

grain. Most importantly, for twins belonging to this subset of ATPs, a twin and its originating GB can be linked. How each ATP formed, by (1) or (2), is not possible to know, but, nonetheless, occurrence of an ATP can be associated with a TT event. The continuation of the ATPs below the EBSD scan surface cannot be guaranteed, and performing serial sectioning would be an appropriate way to confirm it. However, if ATPs were one-dimensional features, the chance that a section of the deformed sample would cut across one such trace intersection would be exceedingly small. On the other hand, previous experimental observation[7] shows that the frequency of observing ATPs in a particular surface is quite high.

Figure 2 shows the variation in the fraction of crossed boundaries with GB misorientation for both Mg and Zr at 3% and 10% compressive strain, respectively. In both cases, a good fraction of {10–12} twins were identified as transmitted twins[7]. The statistics were also sufficiently large to obtain ATPs over a wide range of misorientation. Without a doubt, the results indicate that for both metals, the propensity for ATPs strongly depends on GB-misorientation angle. The relationship is non-linear, where the likelihood for TT is high at low-misorientation angles and drops rapidly once the misorientation angle reaches ~50° for Mg and 60° for Zr. These angles are cutoff misorientation angles above which TTs are rarely observed.

While the trend in Fig. 2 has already been reported previously for Mg[7], here we show that despite the difference in microstructural properties and deformation conditions, it is also present in Zr. More interesting are a few subtle differences. First, we see that TT across low GB misorientation occurs more often in Zr than in Mg. Also, the angle above which TT is substantially suppressed is higher in Zr than in Mg (60° versus 50°). Taken together, it appears that TT is favoured in Zr over Mg, despite the fact that the average twin thickness and twin volume fraction is greater in Mg compared to Zr.

**TT measure.** The results above clearly indicate that the misorientation angle between the neighbouring grains affects TT. The cause has been commonly attributed to the increased misalignment in the twin plane and shear direction when crossing the GB. More specifically, over the years, the likelihood of slip-slip, slip-twin and twin-twin transmissions across grain

boundaries has been characterized via the following geometric measure[6,15–17,19,20]

$$m' = \left( \hat{\boldsymbol{b}}^{(1)} . \hat{\boldsymbol{b}}^{(2)} \right) \left( \hat{\boldsymbol{n}}^{(1)} . \hat{\boldsymbol{n}}^{(2)} \right) \qquad (1)$$

for the (mis)alignment of the glide or twin plane normals and shear directions of the incoming and outgoing systems. In equation (1), the plane normals and shear directions are unit vectors.

Related to $m'$ is a second factor $m'' = \hat{\boldsymbol{b}}^{(1)} . \hat{\boldsymbol{b}}^{(2)}$, which only takes into account the alignment of the incoming and outgoing twin shear vectors. This factor implies glide planes or their intersection trace with the GB play a small or negligible role compared to the direction of shearing. A third and final geometric factor that has been considered in conjunction with $m'$ is the macroscopic Schmid factor (MSF) for the (possible) outgoing twin $m_0$. $m_0$ would represent the notion that TT is independent of the GB misorientation and is a consequence of the neighbouring grain orientation with respect to the macroscopic loading state. The rationale for the latest criterion is that if the parent grain with the incoming twin of {10–12} type were well suited for twinning under the current loading, any neighbouring grain that deviates in orientation from the parent would be less favourable for twinning. Consequently, the further the neighbour deviates from the parent, the less likely it is to twin.

To compare these three geometric factors, a simple bicrystal set-up is considered (shown in Fig. 3a). The orientation of the grain with the incoming twin is fixed and its $c$ axis is normal to the imposed compression. It corresponds to ~0.5 Schmid factor for the incoming twin. A total of 221 different orientations are considered for the neighbouring grain. All three geometric measures are calculated for all the twin variants of the neighbouring grain orientations with respect to the loading direction (for $m_0$) and/or the incoming twin (for $m'$ and $m''$), and the maximum geometric measure, which corresponds to the best-oriented twin variant, is selected. The measure $m'$ and $m''$ range from 0 to 1, and the MSF $m_0$ range from 0 to 0.5. To have the same range as $m'$ and $m''$, we used $2m_0$ in this comparison.

Figure 3b maps these geometric indicators together: $m'$, $m''$ and $2m_0$ for Mg. The map for Zr is only slightly different due to $c/a$ ratio. Generally, all three measures are found to provide the right trend, that is, the propensity for TT decreases with increase in

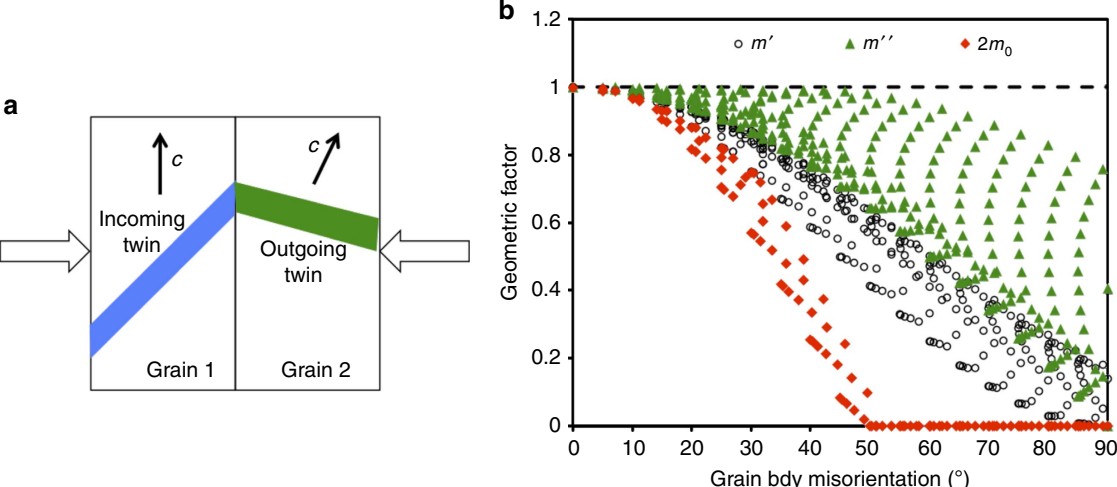

**Figure 3 | Characterization of twin transmission using geometric measures.** (**a**) Schematic representation of the grains with incoming and outgoing twins with loading direction. (**b**) Geometric factor for twin transmission across grain boundaries for Mg. Three geometric factors are considered: $m'$, $m''$, $2m_0$. The $m'$ quantifies the alignment of the shear direction and plane normal of the incoming and best oriented outgoing twins and it is defined in equation (1). The $m''$ only considers the alignment of the shear directions of incoming and best oriented outgoing twins. Finally, $2m_0$ represents twice the Schmid factor of the best oriented twin systems in the neighbouring grain (note: it can be shown that the map for Zr is similar). Note that the Schmid factor for the larger misorientation cases ($> \sim 50°$) can be negative, in which case we assign a zero value to it, to consider only the forward twin shear direction.

misorientation angle. For low-misorientation angles, they all appear to be successful since qualitatively the geometry factors are high for misorientations where transmitted twins are seen. However, none of the indicators provides a good match to the data for the full misorientation angle range. In the case of $m'$ and $m''$, the decline in transmission likelihood is gentler than that seen experimentally. They both overestimate the likelihood of transmission at mid- and high values of misorientation angles and lack a well-defined cutoff misorientation angle for TT. Only for low angles, where transmission is witnessed most often, $m''$ would appear to be in good agreement with observation.

The MSF in the neighbouring grain $2m_0$, which disregards GB misorientation, provides the closest result to the data. Consistent with the data, a well-defined cutoff angle below which transmission is not expected manifests in $m_0$. Because of the influence of $c/a$ ratio on the twin boundary inclination, the cutoff for Zr is slightly higher than Mg (49.1° versus 48.9°). From the qualitative agreement, it might be concluded that it is not the high-misorientation angle that suppresses TT, but the fact that the neighbouring grain becomes poorly oriented for twinning as it deviates further from the parent, which is suitably oriented for twinning. However, the difference in cutoff angle seen experimentally is much greater (50° versus 60°).

As mentioned earlier, these geometric criteria are only partial indicators for TT because they do not include material aspects like elastic moduli, critical resolve shear stress (CRSS) values of plastic deformation modes and $c/a$ ratio. A physical variable that may be a key is the crystallographic slip. In general, when an hcp crystal is deformed, slip occurs alone or together with twinning. For instance, the characteristic twin shear for the most frequently activated twin {1012} ranges from 12.6 to 17% for Mg and Ti. Even if a twin lamella were to overtake the entire grain, it still could not accommodate all of the strain in the grain and slip must occur simultaneously.

**Crystal plasticity modelling**. To investigate the role of plasticity on TT, we need a 3D full-field mechanics model that accounts for anisotropic elasticity, crystal plasticity and discrete twin lamellae within an explicit representation of grain misorientation. Here we employ a spatially resolved crystal plasticity fast Fourier

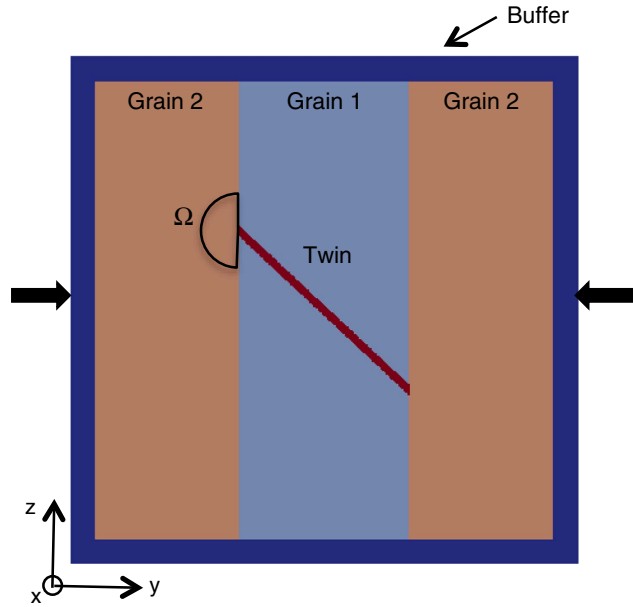

**Figure 4 | Fast Fourier transform model set-up.** Schematic representation of the tricrystal unit cell. The twin is embedded in a central grain (grain 1) and the twin front is arrested at the grain boundary with the neighbouring grain (grain 2). A buffer layer of random orientations surrounds the tricrystal. This configuration represents a polycrystal in which only three grains are magnified for numerical simulation purposes. The region $\Omega$ in the neighbouring grain is the potential region where twin transmission may occur and where the stresses available for twin transmission are calculated.

transform (CP-FFT) technique[25] to calculate the stress fields in a neighbouring grain produced by an impinging twin lamella at a GB (details in Supplementary Note 1).

Figure 4 shows the model of a 3D tricrystal, which consists of a central grain flanked by two equal-sized neighbouring grains with the same orientation. An outer buffer layer of randomly oriented polycrystal material surrounds the tricrystal and represents the

| Table 1 | Elastic and plastic parameters of hcp Mg, Zr and Ti. | | | | | | | | |
|---|---|---|---|---|---|---|---|---|---|
| Material | Temp | Elastic constants (GPa) | | | | | CRSS values of slip modes (MPa) | | |
| | | C11 | C12 | C13 | C33 | C44 | Basal $<a>$ | Prismatic $<a>$ | Pyramidal $<c+a>$ |
| Mg | RT | 58.58 | 25.02 | 20.79 | 61.11 | 16.58 | 3.3 | 35.7 | 86.2 |
| Zr | 76 K | 154.2 | 67.80 | 64.80 | 171.6 | 35.80 | 700.0 | 20.0 | 160.0 |
| Ti | 298 K | 162.4 | 92.00 | 69.00 | 180.7 | 46.70 | 120.0 | 60.0 | 180.0 |

RT, room temperature.
Elastic constants for Mg[30], Zr[31] and Ti[31] are taken from Simmons and Wang[32] and CRSS values for different plastic slip modes used in the calculation for Mg[28], Zr[29] and Ti[33,34]. For Zr, the basal slip system is not activated at all in the present work and in the referenced work[29]. So, the high CRSS for basal slip is equivalent to not accounting for this mode.

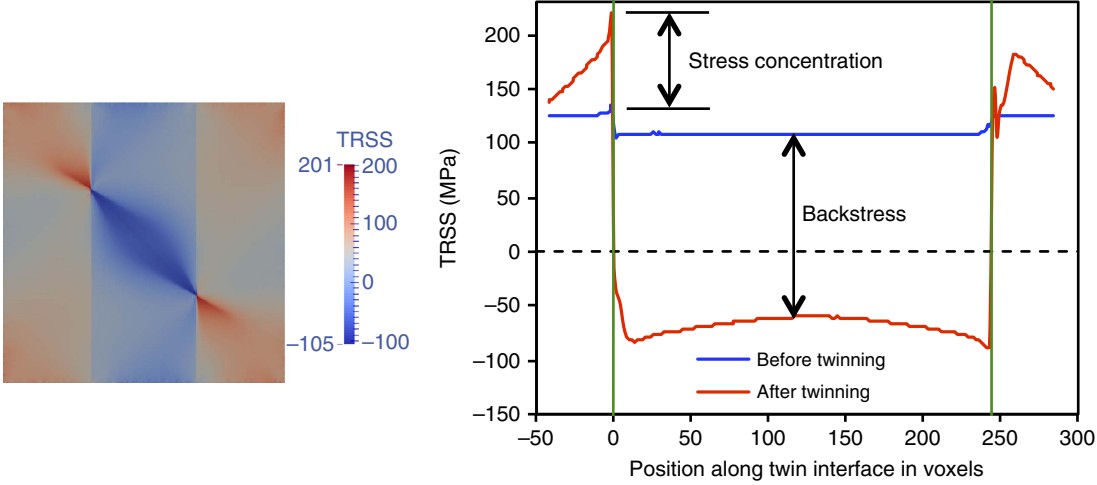

**Figure 5 | Distribution of twinning shear-induced TRSS field.** TRSS field after twinning and the TRSS profile along twin boundary before and after twinning in Zr for the neighbour orientation (0°, 30°, 0°). The stress concentration ahead of twin tip is marked. The twin tip/grain boundary junctions are marked by two vertical green lines.

bulk material response. The $c$ axis of the parent grain is oriented along the $z$-direction, that is, (0°, 0°, 0°). Its twin corresponds to the variant $(01-12)[0-111]$, which is the preferred variant under the applied state of strain. The inclination of the twin boundary with respect to the $y$-direction is 43.1° and 42.6° for Mg and Zr, respectively.

The constitutive law used in the model is based on an infinitesimal elastoviscoplastic FFT formulation, allowing for elastic anisotropy and crystallographic slip on multiple slip families and thus is well suited for hcp crystals[26]. It applies to all material points in the tricrystal: the twin lamella, parent grain and neighbouring grains. Elastically both Mg and Zr are anisotropic and their elastic constants are given in Table 1. The anisotropy indices for Mg of 1.09, 1.01 and 1.23, and for Zr of 0.78, 1.21 and 1.42, where 1.0 represents isotropy[27] indicate that, by all measures, Zr is elastically more anisotropic than Mg. In the simulations we assumed no work hardening and kept the CRSS values constant. The rationale behind this assumption is that, qualitatively, the ratios of CRSSs between soft and hard slip systems are expected to remain constant. As a consequence, it is mostly the orientation of the soft or hard systems in the neighbouring grain that matter.

Independent of their elastic anisotropy, the plastic deformation of hcp crystals, such as Mg and Zr, is anisotropic and carried by multiple slip systems including basal $<a>$, prismatic $<a>$ and pyramidal $<c+a>$ slip[2] with significantly different activation stresses. This leads to anisotropic plastic deformation behaviour for single crystals and for textured polycrystals. The initial CRSS for these slip families

have been characterized previously[28,29] and their values, corresponding to the deformation temperatures of Mg (300 K) and Zr (76 K), are provided in Table 1.

Every simulation involves a series of steps to best replicate the deformation history leading to a twin. First, the tricrystal is strained so that the CRSS of the twin is reached in the parent crystal. A compressive strain in the $y$-direction is applied at a rate of $10^{-3}$ per s to the assembly such that a stress state is generated in the grain that would in principle produce a twin. This state of deformation is mediated by a combination of elasticity and plasticity. Without the twin, the stress field under an applied strain is nearly uniform. The corresponding component of stress resolved on the twin plane and in the twin direction of the $(01-12)[0-111]$ twin, called hereinafter the twin resolved shear stress (TRSS), exceeds the characteristic CRSS for twinning, 20 MPa for Mg[28] and 102 MPa for Zr[29]. Under this deformed state, a single twin lamella is inserted with the characteristic twin shear (12.9% for Mg and 16.6% for Zr) and lattice orientation relationship with the parent, with both twin tips terminating at the two vertical grain boundaries. The methodology for imposing the twin reorientation and shear is described in ref. 25.

Figure 5a shows the TRSS stress field that develops in Zr after twinning under the current applied strain when the neighbouring grain orientation is (0°, 30°, 0°). Figure 5b presents the TRSS profile along the twin boundary before and after twinning. Before twinning, the TRSS in the twin lamellae is homogeneous and ~108 MPa, which is greater than the CRSS (102 MPa). When the twin forms, the shape change caused by the twin shear is constrained by the neighbouring grains. Consequently, a localized

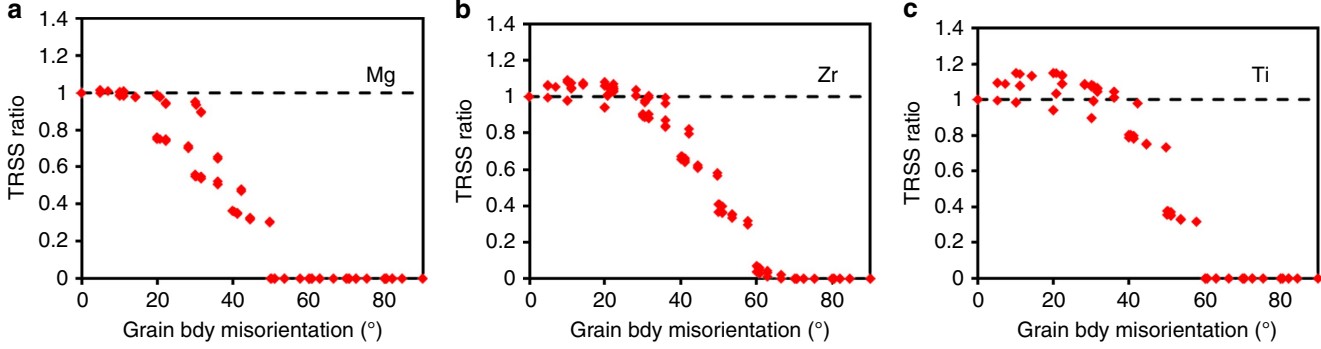

**Figure 6 | FFT model-based evolution of twin transmission indicator.** Evolution of the TRSS ratio ($TRSS_{tip}/TRSS_{SX}$) in region $\Omega$ with grain boundary misorientation angle for (**a**) Mg, (**b**) Zr and (**c**) Ti.

strain field is created around the terminating twin tip in the neighbouring grain, which is indicated by the stress concentration in Fig. 5b. Far from the tip/GB junction, however, the deformation state recovers the uniform field in that grain. The local slip at the twin/GB junction is inhomogeneous and accommodated by different slip activity, 8.2% by prismatic $<a>$, 91.8% by pyramidal $<c+a>$ and no basal $<a>$ contribution. This slip activity is different than far away from the twin in the parent grain and in the neighbouring grain. While challenging to validate directly, this prediction generally agrees with TEM studies on terminating twin tips that find emissary dislocations of both $<a>$ and $<c+a>$ types[35,36].

Most importantly, this localized shear strain field produced at the twin/GB junction is sufficiently high that it could potentially cause a twin to form on the other side of the GB into the neighbouring grain. While it is possible to have homogeneous slip and/or twin nucleation at an arbitrary location in the neighbouring grain, the local stress concentration generated at the twin tip is expected to promote twin nucleation in the vicinity of the tip in the neighbouring grain. Using the TRSS as an acceptable measure of the driving forces needed to propagate the twin forward, we calculated the maximum TRSS among all six twin variants of the neighbouring grain. Calculation of the TRSS naturally takes into account the misorientation between twin planes and Burgers vectors and differences in crystallographic slip across the GB. Driving forces other than the TRSS may be relevant for propagating twins and will be addressed later.

**GB misorientation effects on TT**. To investigate the effects of grain neighbour on the driving forces for TT, the tricrystal deformation calculations are repeated with the crystallography of the parent and twin fixed, but changing the orientation of the two flanking neighbouring grains over the entire orientation space. Using small angular increments of ~2°, the number of distinct neighbour orientations considered in the simulations is 221. In these cases, the twin tips terminate at the grain boundaries. To quantify the driving force for TT, we calculate the average TRSS, called $TRSS_{tip}$, in the stress concentrated volume $\Omega$ ahead of the twin tip (see Fig. 4). For all GB misorientations, we use the same size $\Omega$ but choose the twin variant in the neighbouring grain with the highest TRSS among the six possible. Further, we need a reference state for which twin propagation is certain. The natural one is the case of zero misorientation between the propagating twin and its neighbouring grain. For this case, the orientation of the entire tricrystal equals that of the parent (0°, 0°, 0°) and the ends of the twin lamella correspond to twin tips that terminate within the parent grain. In this special situation, we found that the twin-tip stress concentration $TRSS_{SX}$ drives twin propagation within its own crystal without a need for an increase in the

applied driving force, which can be expected. For Zr, we calculate $TRSS_{SX}$ to be 124.8 MPa and for Mg it is 23.3 MPa. The reasons for the difference are many: differing elastic constants, CRSS values for slip modes, twin shear, twin-matrix orientation relationship and applied strain in order for the TRSS to reach the twin CRSS.

Figure 6 maps the TRSS ratio ($= TRSS_{tip}/TRSS_{SX}$) with the corresponding grain neighbour misorientation angle. Values of $TRSS_{tip}$ close to $TRSS_{SX}$ can be associated with high chances for transmission. Normalization of $TRSS_{tip}$ with $TRSS_{SX}$ helps to nullify the effect of differences in the characteristic twin shear among hcp materials. For both Mg (Fig. 6a) and Zr (Fig. 6b), the TRSS ratio decreases with misorientation angle, which would indicate a reduction in the propensity for TT into the neighbouring grain with misorientation. The most important aspect of Fig. 6 is that the computed TRSS ratio follows remarkably well the experimentally measured variation in TT occurrence with misorientation angle (Fig. 2). In particular, the calculation captures the fact that this reduction is more pronounced in Mg than in Zr. The model even forecasts, consistent with experimental observations, that the cutoff misorientation angle is ~50° for Mg and 66° for Zr. We note that the EBSD statistical information displayed in Fig. 2 indicates a small number of ATPs in Zr at GBs with misorientation angles larger than this cutoff. We attribute the discrepancy to possible large stress deviations associated with clusters of small-sized Zr grains, where twin variants with negative 'macroscopic' Schmid factor are activated. Such deviations are not generated in the model tricrystal.

A noteworthy event emerges from the calculations for Zr. For some grain neighbours with low-misorientation angle, the propensity for propagating the twin tip is even higher than that for a single crystal with no misorientation angle (TRSS ratio $>1$). This finding implies that for these special grain neighbours in Zr, the driving force for propagating the twin across the GB is even higher than that for propagating the same twin within its parent grain. This TT 'boost' is not predicted for Mg and this difference provides another indication that plastic deformation properties matter.

Not all low-misorientation angle GBs create this boost in Zr. Therefore, obtaining experimental validation for these special cases would not be straightforward. Nonetheless, their existence in Zr, but not in Mg, is consistent with the experimentally observed trend that, particularly at low misorientation angles, Zr is more likely to transmit than Mg.

Such critical material differences would not have been predicted by geometry only, using for an instance a factor that accounts for the misalignment in the twin plane and twinning direction orientation across the GB (Fig. 3). While the analysis based on the third geometric measure (i.e., $m_0$, the MSF of the

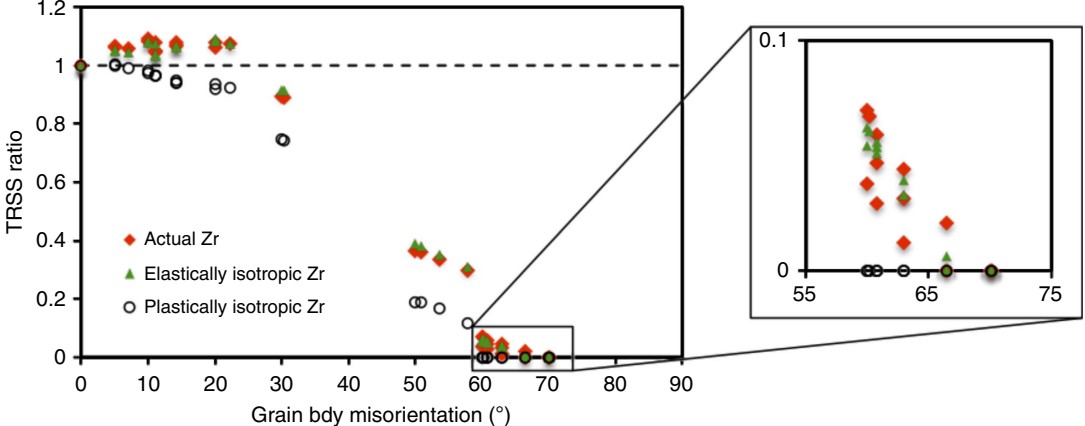

**Figure 7 | Role of plastic anisotropy on ATP formation.** Mapping of TRSS ratio in the region Ω as a function of neighbour grain boundary misorientation, for elastically isotropic Zr, plastically isotropic Zr and actual Zr.

neighbouring grain) gives a similar trend for ATP formation as the present FFT-based predictions, it fails to account for critical material differences that also affect ATP formation. The calculations reveal that the localized plastic response at the twin-tip/GB junction drives TT. This field is not only a consequence of the slip properties and orientation of the two adjacent crystals but it also depends on the differences in PA. In light of the outstanding difference between the PA of Mg and Zr, it is only natural to postulate that the twin induced stresses and PA (their dependence on orientation) must affect TT. In support, we carry out two sets of calculations, one involving a hypothetical Zr material where either the elastic or the plastic properties were made isotropic, and another for Ti, which is less plastically anisotropic than Zr but more so than Mg.

To create an elastically isotropic Zr, we modified the elastic constants of Zr without changing the activation stresses of its plastic slip modes. To do so, only the elastic constants $C_{33}$ and $C_{44}$ are altered from 164.9 and 32.1 to 145.0 and 35.0 GPa, respectively, making sure that we do not violate the Cauchy conditions. The resulting anisotropic indices are 1.08, 1.01 and 1.16, which are close to Mg. Likewise, to make Zr plastically isotropic we modify the activation stresses of the different slip modes without changing the elastic constants. The activation stresses for basal and pyramidal slip changed from 700 and 160 MPa to 100 and 60 MPa. We perform simulations for 35 different neighbouring orientations making sure to include low-misorientation-angle cases where the TT boost was predicted and high-misorientation-angle cases that are close to the cutoff angle.

The TRSS ratios for the two 'designed' Zr materials are shown in Fig. 7. The calculations indicate that PA, not elastic anisotropy, is responsible for the TT boost at low-misorientation angles and the value of the cutoff misorientation angle. As shown, the nearly elastically isotropic Zr material behaves very similarly to the actual Zr material. For plastically isotropic Zr, the TRSS ratio no longer exceeds 1.0 for any neighbouring orientation. The cutoff misorientation angle for plastically isotropic Zr is 60°, which is smaller than the cutoff angle for actual Zr ($\sim$66°). By reducing the PA, the probability for TT is reduced.

To confirm the effects of PA revealed above from studies of a 'designed' material, we perform a set of calculations on Ti. TT has been reported prevalent in other hcp polycrystals and, in particular, Ti and its alloys[19,20]. Table 1 presents the elastic constants for pure Ti. With these, we calculate the elastic anisotropic indices to be 1.06, 0.75 and 1.21, indicating that Ti is more elastically anisotropic than Mg but less so than Zr. Like Mg, all three-slip families (prismatic $<a>$, basal $<a>$ and

pyramidal $<c+a>$) are active in the plastic deformation of Ti[33,34]. The corresponding activation stresses for these slip modes in Ti at room temperature are given in Table 1. Based on the number of available slip modes and the relative differences between their activation stresses, Ti can be considered plastically more anisotropic than Mg but less so than Zr. Based on our hypothesis and the foregoing analysis on the designed Zr, it would be expected that TT in Ti would be more likely than in Mg but less likely than in Zr.

Figure 6c shows the driving force TRSS ratio for TT for the same broad span of 221 grain neighbour orientations as considered earlier for Mg and Zr. For Ti, $TRSS_{SX} = 127.3$ MPa. As in Mg and Zr, TT propensity decreases with increasing misorientation angle. This prediction is in good agreement with the EBSD-based experimental results reported for pure Ti and Ti alloys[19,20].

Referring to calculations shown in Fig. 6, we found that compared to Mg, TT is more likely in Ti. Compared to Zr, there are many more low-misorientation neighbours for which the driving force exceeds 1.0. Thus, even more special grain neighbour orientations exist for which the driving forces for TT are higher than those to propagate the same twin within its parent grain. This difference is likely an outcome of the higher characteristic twin shear for Ti than Zr (17.4% compared to 16.6%): for the same size twin, the amount of plastic strain ahead of the twin tip is greater in Ti. We also observe in Fig. 6c that the cutoff angle for Ti is predicted to be close to 60°, which is greater than that of Mg ($\sim$50°) and less than that of Zr ($\sim$66°). Taken together, these results support the general notion that a greater probability for TT can be expected with a greater degree of PA.

Driving forces associated with atomic-scale processes of TT are not taken into account in our calculations. Twins impinge into a GB and in order for TT to occur, twin nuclei need to form and emanate into the neighbouring grain, possibly altering the GB in the process[37]. The driving forces for these processes may differ from the TRSS used here. Should they prove later to be relevant, driving forces associated with other mechanisms of twin boundary migration could be extracted from the present calculated fields. Nonetheless, good agreement on several aspects between our calculations and the experimental data for both Mg and Zr provides compelling evidence that PA is a key variable governing TT.

The chief result of our analysis is that, with all else being the same, increasing the degree of PA promotes TT. This result may appear contradictory to current explanations for TT that do not include the characteristics of material behaviour, only

crystallography, such as the orientation relationship between the twin variants on both sides of the GB. However, PA is a material property that couples plastic behaviour with crystallography. Therefore, the finding that PA has such a profound influence on the propensity for TT is, in fact, related to conventional ideas on crystal geometry effects. It, however, provides much greater insight than just geometry arguments alone. For instance, it implies that with all else being the same—texture, GB-misorientation distribution and loading state—the probability of TT can be reduced under deformation conditions or material compositions that reduce its PA. The PA of an hcp material is related to the differences in the critical resolved shear stresses for activating different modes of slip, such as basal $<a>$, prismatic $<a>$ and pyramidal $<c+a>$ slip. Experimental evidence indicates that these activation barriers can be altered via alloying additions[38,39], changes in applied strain rate and deformation temperature[29].

In summary, an automated EBSD method is used to map statistically significant numbers of {1012} TT events in GB misorientation space for both pure polycrystalline Mg and Zr. The results show that the tendency for TT decreases with an increase in GB misorientation angle for both Mg and Zr. However, it is found that Mg and Zr differ significantly in their propensity for TT, particularly across grain boundaries with low-misorientation angles. The analysis reveals the existence of an upper cutoff misorientation angle above which TT is rare and a significant 10° difference between the cutoff angles for Mg and Zr. To understand the origin of the those material effects on TT not associated with pure geometrical considerations, a 3D full-field crystal plasticity-based fast Fourier transform technique is employed to study the driving forces underlying TT. The results achieve good agreement with the data unlike conventional measures based on geometry alone. Most significantly, the analysis shows that PA is a critical factor that governs the likelihood of TT. With all else being the same, increasing PA promotes TT. These results can help guide alloy selection to reduce the propensity of TT and the potential instabilities it may cause.

**Data availability**. The data that support the findings of this study are available from the corresponding author upon reasonable request.

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

## Acknowledgements

The authors are grateful to Dr Ricardo Lebensohn for making available the FFT-EVPSC code used here for the simulations. This work is fully funded by the U.S. Department of Energy, Office of Basic Energy Sciences Project FWP 06SCPE401.

## Author contributions

M.A.K. performed the crystal-plasticity calculations, M.A.K., I.J.B. and C.N.T. analysed the results. R.J.M. performed the EBSD-based statistical analysis. All authors contributed to the writing of the manuscript.

## Additional information

**Competing financial interests:** The authors declare no competing financial interests.

