## [Peer Review File · Nature Communications]

Reviewers' comments:

Reviewer #1 (Remarks to the Author):

This paper is a lovely piece of work that describes correlations between neighbour grains, slip and twinning.

The authors focus on plastic anisotropy and its effect on enabling twin propagation. However they omit discussion that plastic anisotropy, or rather the competitive ability of twinning to carry strain in grains of different orientations is, in addition to difficulties in twin nucleation (which in effect drives twin-twin deformation cascades) will be responsible for the effects correlated within this work. In effect, if plastic slip is homogeneous and easy (relative to twin nucleation), then twinning will not readily occur. Twin nucleation is difficult, and requires a local stress riser. In heterogeneous twin nucleation this can be provided, for instance, by a prior twin terminating near a grain boundary or slip stimulated twin nucleation. However if the strain in the neighbour grain can be accommodate easily through slip (e.g. more homogeneous slip) then cascades of twins are less likely to occur. It would be worth drawing the issue of competitive accommodation of deformation during loading and the role of nucleation vs growth vs slip out more in the discussion.

In the introduction the authors comment that: "Twin lamellae, however, do not necessarily remain within the original parent grain under continued straining, and provided the conditions are right, can propagate across the grain boundary into the neighbouring grain. They can continue to transmit across grain boundaries, hence percolating across the grain structure, forming so called twin chains or catalytic twins^{4, 6-7}."

This is not strictly correct. It is unlikely/uncertain (and there is no direct evidence) that twins can perform a direct transfer in the majority of cases. Indirect mechanisms may operate, leading to a cascade of twin assisted twin nucleation which results in continued propagation of shear through twinning in a twin chain. This subtlety is an important issue in twin-mechanics and would be worth clarifying by the authors.

For equation 1 - I think that these should likely to be unit vectors (or else the alignment considers the magnitude of the strain which opens up a separate line of discussion). This is similar for m' .

Utility of the macroscopic Schmid factor for ATP is very interesting. Its ability to predict the likelihood of twin linking indicates that the second grain was likely to twin anyway, but that the initial twin provides a small extra driving force locally at the grain boundary to enable indirect/direct twin transmission across the interface.

Table 1 - the elastic constants need referencing properly. There are multiple values / referencing for these constants and ideally these should be references against the original papers (with a

reference to Simons and Wang if the authors' choose). It would also be useful to include a temperature column (for aid of the reader).

Table 1 - the CRSS values for Zr are strange (I know these are based upon previous fitting of data). However looking at the graphs in Figure 2 of for 76K deformation of Zr with IPC and IPT (and given the initial texture of Zr), the fitted flow stresses are unlikely in a VPSC model to depend strongly on both

The authors need to introduce the EVP-FFT constitutive relationship (in brief) and also the chosen hardening scheme. This has some important implications on how to interpret / use the CRSS values in future work (CRSS from a power law based model are not directly transferable into a physically based backstress model for instance). How do the authors tackle issues of spectral leakage (due to elastic and plastic contrast) in the iterative scheme? Does this have an impact on the case of elastic isotropy vs elastic anisotropy?

Minor comment - prior work by this author group on low temperature deformation of Zr reports that the deformation was at 76K rather than 77K as reported here.

Figure 1 caption - please include the strain rate.

Reviewer #2 (Remarks to the Author):

Overall, this is a very nice manuscript that provides additional understanding to the observation that in many hexagonal metals deformation twinning can propagate from grain to grain, often in what appear to be cascades of twins. The large data sets presented here, facilitated by an automated EBSD technique, allow a statistical approach to be brought to bear on the problem. The highlight of the work is the CP-FFT analysis that has led to the conclusion that the plastic anisotropy, rather than elastic anisotropy, is the critical factor in driving this twin-to-twin nucleation. Overall, I find the work to be original and of interest to the community.

While I do really like the work, I did at times find the manuscript somewhat unclear and difficult to follow. Some of these issues are outlined below, but I also think it would be beneficial if the authors ensure that consistent terminology is used throughout the manuscript. My specific comments follow, in no particular order.

Early in the manuscript it is stated, "Twin lamellae, however, do not necessarily remain within the original parent grain under continued straining, and provided the conditions are right, can propagate across the grain boundary into the neighboring grain". Despite my familiarity with the process being studied in this manuscript, initially I was confused by this statement. In fact, the same twin does not propagate in to the next grain (with the exception of the offset of the grain boundary that results from the twin shear), but instead a corresponding twin (in a twin orientation with the neighboring grain and with some disorientation to the original twin) nucleates and

grows in the neighboring grain. So I suggest the first paragraph be re-worked a little to clarify the process.

With a density of 6.49 g/cc, is Zr considered for light-weight structural applications? I typically think of Zr being used for corrosive environments.

Considering the twin-twin phenomenon as I understand it, I have always been left with the question, which is reinforced by my reading of the manuscript: If the twins are correlated (defined as adjoined twin pairs - ATPs in the manuscript note, that it might be beneficial to introduce this term earlier in the manuscript) at the polished surface of the samples, do the authors, or anyone else, know if these same ATPs are also correlated below the surface? If they are, one of two cases must exist. The first is that the twin plane intersections with the grain boundaries must form a common line for both the original and second twin (which would be the case for tilt boundaries, but not for boundaries that have a twist component). The second is that either or both of the twins must have a variable or stepped twin plane with its matrix so that the twins are not only aligned at the surface, but also below the surface (perhaps forming by cross-slip of the twinning partial dislocations, perhaps by nucleating a series of twinning partials from the grain boundary, perhaps by some grain boundary accommodation mechanism). The first seems like an unlikely condition, which would only lead to the correlated twinning under very specific conditions of grain disorientation and boundary orientation, while the second seems mechanistically complicated and perhaps energetically unfavorable. Along these lines, the twins in Mg (figure 1) appear to be very wavy, suggesting a lot of dislocations (and steps) in the twin boundaries and that forming steps on the twin boundaries to meet this condition would not be difficult. But in Zr, while some twins are lenticular, many of the ATPs are very flat, suggesting it might be more energetically dis-favorable to form steps in the twin boundaries. A related question is if the deformed structures have a certain fraction of twins, what is the statistical likelihood that a twin in one grain will line up at a boundary with another twin in the neighboring grain in some random surface section? Have the authors (or anyone else) ever done any serial polishing to determine if the same twins remain correlated over some depth of crystal? I am not suggesting that the correlated twinning process does not happen, but instead trying to understand how it happens.

I am rather confused by a number of things in figure 3. First, is the data plotted only for cases where twin transmission has been observed (assumed, but not stated anywhere)? Second, the factor m_o is defined as "the Schmid factor for the (possible) outgoing twin". Presumably, this Schmid factor is a global Schmid factor, based on the overall state of stress and not the local state of stress, and presumably is it based on uniaxial compression (the nature of the compression described in section 2.1 is not entirely clear). Given this, m_o cannot exceed 0.5 (the sum of two cosines, restricted by the slip plane normal and slip direction being at 90 degrees to each other), but figure 3 shows it ranging between 0 and 1. Has this Schmid factor been normalized in some manner? Third, it is not clear why m_o goes to zero at misorientation angles greater than about 49 degrees. There is a rationalization given that this cut-off is different for Mg and Zr because of differences in c/a ratios, but the rationalization is not clear. Is there an underlying assumption about the orientation of the parent grain that leads to this cut-off angle?

In the second paragraph of page 8, it is stated "The initial activation stresses for these slip families". This "initial activation stress" is typically referred to the critical resolved shear

stress (CRSS), so why not use it here. After all, the abbreviation CRSS is used later in the manuscript, but it has not been defined.

Does the model incorporate the crystallographically imposed twinning shear?

Is the stress that develops in the simulation, shown in figure 5, periodic in the direction perpendicular to the page (i.e. in the x-direction shown in figure 4)? If so, is this the situation in all simulations? If it is, are the simulations limited to case #1 described above, in which the twin planes in both grains share a common intersection with the boundary? If it is not the case, is the stress developed a function of x-direction (i.e. case #2 above). Or does case #1 versus case #2 not matter as the model does not consider discrete dislocation mechanisms and having a twin boundary plane that is not the same as the twinning plane is not an issue?

Why is TRSS used in some places while in others T-RSS is used in other places.

Reviewer #3 (Remarks to the Author):

This study focus on a model predicting if an adjacent twin pairs would be formed or not. Combining a large data set of experimental results get by an automated method and a crystal plasticity model, authors compare different HCP materials (mainly Mg and Zr) to show that the transmission criterion depends not only on the misorientation, but also on intrinsic material plastic properties such as the CRSS of slip and twin systems.

This study is original and interesting, data and methodology are good, uncertainties explained, conclusion is reliable (except the term 'plastic anisotropy'), Reference ok and the text is clear, however the present reviewer suggest some minor corrections.

1-Abstract: plastic anisotropy : for the reviewer a material with a random orientation is a material plastically isotropic, whereas the meaning of the authors is much different, so I suggest to find a better expression to refer to the plastic properties of grains and singlecrystals.

2-introduction p2: 'forming so called twin chains or catalytic twins' : we can also more simply say cross boundary twin or adjacent twin pair [M.R. Barnett, M.D. Nave, A. Ghaderi, Acta Materialia 60 (2012) 1433-1443] or adjoined twin pair as you do later.

3-Figure 1: there is not the direction on the IPF coloring (we guess red is ND), nor the ND and compression direction on the EBSD maps.

4-p6 section 2.2: 'The rationale for the (Schmid factor) criterion is that if the parent grain with the incoming twin is well suited for twinning under the current loading, any neighboring grain that deviates in orientation from the parent would be less favorable for twinning. Consequently, the further the neighbor deviates from the parent, the less likely it is to twin.' The reviewer disagrees with this explanation: If a parent grain have a SF of 0.5 for 1 twin variant, it could have a neighbor grain (with another orientation) with a SF=0.5 for another twin variant. And this other

twin, certainly with a very different plane and shear direction, could join the first twin at the boundary on the EBSD map by chance. In compression this could happen with grains of orientation (0 90 30) and (90 90 30) which are missoriented 90 degree but the SF of the 2 best variant is 0.49.

5- Figure 3: When we read the manuscript and the description, we expect that each point stand for 1 observed ATP, and so are experimental results. But clearly this figure is made with simulations, this is why the points form lines. Also the authors must explain in the text for one pair, how do chose the outgoing twin to compute the Schmid Factor, how do they have a SF m0 between 1 and 0?

6- p7 section 2.2: these geometric indicators (...) do not include materials aspect': for the reviewer's point of view, GB missorientation, grain orientation, twin variant's b and n are all 'material aspects'.

7- p9 section 2.3: as far as we understand, the TRSS is the shear stress along the twin plan and with the twin shear direction in the central grain, but in the 2 neighbor grains, does the shear stress is computed along the same plan which is (usually) not a twin plan or on the best twin variant ?

8-p11 section2.4: does a ratio $TRSS_{tip}/TRSS_{sx} > 1$ also fit with a higher SF in propagated twin than in the central crystal of orientation 0 0 0 ?

9-p15 conclusion: experimentally for Zr some high missorientation (fig 2b, 80degree) also witness twin transmission, whereas the present calculations predict no transmission. According to Fig3, the geometric factors always expect twin propagation (they are seldom null), so the developed model predict the same tendency as m0 the SF, but with higher cut-off missorientation angle. So the presented simulations (using the 3-crystal) are good to compare different materials but not predict perfectly the creation of adjacent twins.

Response to Reviewer - 1

This paper is a lovely piece of work that describes correlations between neighbor grains, slip and twinning.

Response: We thank the referee for his/her kind words and for the careful and detailed review of our manuscript. We found the comments valuable toward improving the manuscript. Following his/her suggestions we added in the appropriate places the requested clarifications and discussions.

The authors focus on plastic anisotropy and its effect on enabling twin propagation. However they omit discussion that plastic anisotropy, or rather the competitive ability of twinning to carry strain in grains of different orientations is, in addition to difficulties in twin nucleation (which in effect drives twin-twin deformation cascades) will be responsible for the effects correlated within this work. In effect, if plastic slip is homogeneous and easy (relative to twin nucleation), then twinning will not readily occur. Twin nucleation is difficult, and requires a local stress riser. In heterogeneous twin nucleation this can be provided, for instance, by a prior twin terminating near a grain boundary or slip stimulated twin nucleation. However if the strain in the neighbour grain can be accommodate easily through slip (e.g. more homogeneous slip) then cascades of twins are less likely to occur. It would be worth drawing the issue of competitive accommodation of deformation during loading and the role of nucleation vs growth vs slip out more in the discussion.

Response: We added a clarification. In the simulations, for plasticity, we considered only slip, not twinning to accommodate the plastic deformation. The twinning shear associated with the existing twin is accommodated by a combination of elastic and plastic deformation. More specifically, the stress concentration generated in the vicinity of the twin tip in the neighboring grain (region Ω in Fig. 4) is a result of elasticity and crystal plasticity. We study how this stress concentration can act as a local stress riser to nucleate or transmit a twin in the neighboring grain. At the same time we agree that any further loading can be accommodated through homogenous slip and/or growth of the existing twin and/or nucleation of new twins in the parent grain or in the neighboring grain. Compared to twin nucleation at an arbitrary location, the local stress riser at the twin tip is preferable for twin nucleation. We have mentioned this point in the manuscript in page 10 and it read as,

“While it is possible to have homogeneous slip and/or twin nucleation at an arbitrary location in the neighboring grain, the local stress concentration generated at the twin tip is expected to promote twin nucleation in the vicinity of the tip in the neighboring grain.”

In the introduction the authors comment that: “Twin lamellae, however, do not necessarily remain within the original parent grain under continued straining, and provided the conditions are right, can propagate across the grain boundary into the neighbouring grain. They can continue to transmit across grain boundaries, hence percolating across the grain

structure, forming so called twin chains or catalytic twins^{4, 6-7}." This is not strictly correct. It is unlikely/uncertain (and there is no direct evidence) that twins can perform a direct transfer in the majority of cases. Indirect mechanisms may operate, leading to a cascade of twin assisted twin nucleation which results in continued propagation of shear through twinning in a twin chain. This subtlety is an important issue in twin-mechanics and would be worth clarifying by the authors.

Response: We agree. We have reworded the sentence:

" However, under continued straining and under the right conditions, twin lamellae can stimulate formation of another twin on the other side of a grain boundary (GB), appearing as if the twin has propagated across the boundary. One can envision that this process may continue, triggering twins in their neighbors and creating so called twin chains or catalytic twins across the grain structure^{4, 6-7}."

For equation 1 - I think that these should likely to be unit vectors (or else the alignment considers the magnitude of the strain which opens up a separate line of discussion). This is similar for m'.

Response: We agree. Both the plane normal and shear direction are unit vectors and this point is now addressed in the manuscript, as,

$$m' = (\hat{b}^{(1)} \cdot \hat{b}^{(2)}) (\hat{n}^{(1)} \cdot \hat{n}^{(2)}) \quad (1)$$

for the (mis)alignment of the normal of glide or twin planes and shear directions of the incoming and outgoing systems. In Eq. 1 the plane normals and shear directions are unit vectors.

Utility of the macroscopic Schmid factor for ATP is very interesting. Its ability to predict the likelihood of twin linking indicates that the second grain was likely to twin anyway, but that the initial twin provides a small extra driving force locally at the grain boundary to enable indirect/direct twin transmission across the interface.

Response: We agree and we have included this important point in the manuscript on page 12, and now it reads as,

"While the analysis based on the third geometric measure (i.e., m_0 , the MSF of the neighboring grain) gives a similar trend for ATP formation as the present FFT based predictions, it fails to account for critical material differences that also affect the ATP formation."

Table 1 - the elastic constants need referencing properly. There are multiple values / referencing for these constants and ideally these should be references against the original papers (with a reference to Simons and Wang if the authors' choose). It would also be useful

to include a temperature column (for aid of the reader).

Response: We agree. We have provided the proper references for elastic constants along with temperature on page 9.

Table 1 - the CRSS values for Zr are strange (I know these are based upon previous fitting of data). However looking at the graphs in Figure 2 of for 76K deformation of Zr with IPC and IPT (and given the initial texture of Zr), the fitted flow stresses are unlikely in a VPSC model to depend strongly on both

Response: We believe that the reviewer is asking whether in the cited work the fitted stress-strain responses using the VPSC model depend on temperature and deformation modes. In that cited work the VPSC model parameters were obtained using a hardening law based on the storage of dislocations by thermally activated rate laws and hence, the CRSS values were a function of temperature and strain-rate. In that work, the VPSC model successfully predicted the differences in stress-strain response for IPC, IPT and TTC loading directions. Contrary to what the reviewer claims, there are substantial differences between these three tests.

Concerning the CRSS value for basal slip systems of Zr we agree that it is very high and it may seem strange. However, both in the cited work and in the present work, the basal slip systems are not activated at all. As a consequence, the high CRSS for basal is equivalent to not accounting for this mode. We have mentioned this point in the revised manuscript.

The authors need to introduce the EVP-FFT constitutive relationship (in brief) and also the chosen hardening scheme. This has some important implications on how to interpret / use the CRSS values in future work (CRSS from a power law based model are not directly transferable into a physically based backstress model for instance). How do the authors tackle issues of spectral leakage (due to elastic and plastic contrast) in the iterative scheme? Does this have an impact on the case of elastic isotropy vs elastic anisotropy?

Response: In this work, we have used the power law based flow rule in the infinitesimal elasto-visco-plastic FFT formulation. In the simulations, we do not account for work hardening or lattice rotation due to slip activity. We also do not account for the twin backstress in the flow rule explicitly. We have mentioned these points in the manuscript in page 8, as

"In the simulations we assumed no work hardening and kept the critical resolved shear stress (CRSS) values constant. The rationale behind this assumption is that, qualitatively, the ratios of CRSSs between soft and hard slip systems are expected to remain constant. As a consequence it is mostly the orientation of the soft or hard systems in the neighboring grain that matter."

We believe that the word "spectral leakage" used by the reviewer means the Gibbs oscillations in the FFT calculation by truncating the higher order frequencies. In this work

we do not explicitly account for the Gibbs oscillations induced by the elastic and plastic contrast across different domains. However, we verified that those are minimal in this work. The latter result is in part due to the fact that twin is inclined roughly at 45 degrees with respect to the Fourier grid, and such situation minimizes the local stress oscillations.

Minor comment - prior work by this author group on low temperature deformation of Zr reports that the deformation was at 76K rather than 77K as reported here.

Response: Thanks for pointing this out. It is 76K. We have corrected it.

Figure 1 caption - please include the strain rate.

Response: Thank you. We have included the strain rate (10^{-3} s) in Figure 1.

Response to Reviewer - 2

Overall, this is a very nice manuscript that provides additional understanding to the observation that in many hexagonal metals deformation twinning can propagate from grain to grain, often in what appear to be cascades of twins. The large data sets presented here, facilitated by an automated EBSD technique, allow a statistical approach to be brought to bear on the problem. The highlight of the work is the CP-FFT analysis that has led to the conclusion that the plastic anisotropy, rather than elastic anisotropy, is the critical factor in driving this twin-to-twin nucleation. Overall, I find the work to be original and of interest to the community.

Response: We thank the referee for his/her careful and detailed review of our manuscript and for his/her valuable comments.

While I do really like the work, I did at times find the manuscript somewhat unclear and difficult to follow. Some of these issues are outlined below, but I also think it would be beneficial if the authors ensure that consistent terminology is used throughout the manuscript. My specific comments follow, in no particular order.

Early in the manuscript it is stated, "Twin lamellae, however, do not necessarily remain within the original parent grain under continued straining, and provided the conditions are right, can propagate across the grain boundary into the neighboring grain". Despite my familiarity with the process being studied in this manuscript, initially I was confused by this statement. In fact, the same twin does not propagate in to the next grain (with the exception of the offset of the grain boundary that results from the twin shear), but instead a corresponding twin (in a twin orientation with the neighboring grain and with some disorientation to the original twin) nucleates and grows in the neighboring grain. So I suggest the first paragraph be re-worked a little to clarify the process.

Response: We agree. We have re-written it as:

"However, under continued straining and under the right conditions, twin lamellae can

stimulate formation of another twin on the other side of a grain boundary (GB), appearing as if the twin has propagated across the boundary. One can envision that this process may continue, triggering twins in their neighbors and creating so called twin chains or catalytic twins across the grain structure^{4, 6-7}."

With a density of 6.49 g/cc, is Zr considered for light-weight structural applications? I typically think of Zr being used for corrosive environments.

Response: We agree. We have corrected it in the manuscript page 1.

In abstract: *"Materials with hexagonal close packed crystal structure, like Mg, Ti, and Zr, are being used in transportation, aerospace and nuclear industry, respectively. ~~for lightweight structural applications for transportation, space, and the nuclear industry.~~"*

In the Introduction: *"Hexagonal close packed (hcp) alloys such as Mg, Ti, and Zr alloys are used in transportation (lightweight), aerospace (corrosion resistance and low thermal coefficient) and nuclear (corrosion and radiation resistance) industry, respectively. These alloys ~~of interest in light-weight structural applications, such as Mg, Ti, and Zr alloys, undergo deformation twinning when strained due to the scarcity of slip systems~~¹⁻³."*

Considering the twin-twin phenomenon as I understand it, I have always been left with the question, which is reinforced by my reading of the manuscript: If the twins are correlated (defined as adjoined twin pairs - ATPs in the manuscript note, that it might be beneficial to introduce this term earlier in the manuscript) at the polished surface of the samples, do the authors, or anyone else, know if these same ATPs are also correlated below the surface? If they are, one of two cases must exist. The first is that the twin plane intersections with the grain boundaries must form a common line for both the original and second twin (which would be the case for tilt boundaries, but not for boundaries that have a twist component). The second is that either or both of the twins must have a variable or stepped twin plane with its matrix so that the twins are not only aligned at the surface, but also below the surface (perhaps forming by cross-slip of the twinning partial dislocations, perhaps by nucleating a series of twinning partials from the grain boundary, perhaps by some grain boundary accommodation mechanism). The first seems like an unlikely condition, which would only lead to the correlated twinning under very specific conditions of grain disorientation and boundary orientation, while the second seems mechanistically complicated and perhaps energetically unfavorable. Along these lines, the twins in Mg (figure 1) appear to be very wavy, suggesting a lot of dislocations (and steps) in the twin boundaries and that forming steps on the twin boundaries to meet this condition would not be difficult. But in Zr, while some twins are lenticular, many of the ATPs are very flat, suggesting it might be more energetically dis-favorable to form steps in the twin boundaries. A related question is if the deformed structures have a certain fraction of twins, what is the statistical likelihood that a twin in one grain will line up at a boundary with another twin in the neighboring grain in some random surface section? Have the authors (or anyone else) ever done any serial polishing to determine if the same twins remain correlated over some depth of crystal? I am not suggesting that the correlated twinning process does not happen,

but instead trying to understand how it happens.

Response: We agree. We have now introduced the term ATPs on page 2.

We did not perform serial sectioning of the sample and are unaware of any previous, statistically significant serial sectioning study of this kind that might be used to determine whether the observed ATPs at the polished surface extend below the surface. We agree that to do so would be a good idea. However, we have good reasons to believe that the ATPs continue below the surface for an extended volume and that they are not merely connected by a relatively narrow intersection. Our previous study in Beyerlein et al., 2010, Phil Mag, and the images in Figure 1 show that the twin thickness at grain boundaries is larger for the ATPs than that for isolated twins, and in many cases, the thickness measured in the sample plane for the two ATP twins is practically identical. This supports the idea of induced accommodation of twinning shear and relief of back stresses at the intersection. This also suggests that deviation from the ideal twin relationship as discussed by the reviewer in the second case above to be more likely over an extended volume even for twins that are not well aligned. If the twins met at points at the boundary and were misaligned, it would be more difficult to accept this argument.

I am rather confused by a number of things in figure 3. First, is the data plotted only for cases where twin transmission has been observed (assumed, but not stated anywhere)? Second, the factor m_0 is defined as "the Schmid factor for the (possible) outgoing twin". Presumably, this Schmid factor is a global Schmid factor, based on the overall state of stress and not the local state of stress, and presumably is it based on uniaxial compression (the nature of the compression described in section 2.1 is not entirely clear). Given this, m_0 cannot exceed 0.5 (the sum of two cosines, restricted by the slip plane normal and slip direction being at 90 degrees to each other), but figure 3 shows it ranging between 0 and 1. Has this Schmid factor been normalized in some manner? Third, it is not clear why m_0 goes to zero at misorientation angles greater than about 49 degrees. There is a rationalization given that this cut-off is different for Mg and Zr because of differences in c/a ratios, but the rationalization is not clear. Is there an underlying assumption about the orientation of the parent grain that leads to this cut-off angle?

Response: We thank the reviewer for these points. Below we address them one by one.

The data plotted in Fig. 3 are not from experimental observations. We have assumed possible neighboring grain orientations while fixing the parent grain orientation. And then we have calculated the geometric measures that are shown in Fig. 3. It gives an idea about twin transmission as function of grain boundary mis-orientation and it is not representing the observed ATPs, which are shown in Fig. 2.

The factor m_0 is the macroscopic Schmid factor. It is based on the macroscopic compression direction. We have shown schematically the loading direction and the orientation of both the grains with twin in Fig. 3(a). We agree that m_0 ranges from 0.0 to 0.5. To have a same range for all three measures (m' , m'' and m_0), we have used $2 m_0$ and this point is now mentioned in the text [page 6] and in the Figure 3.

The highest possible Schmid factor for the twin system is 0.5, which corresponds to a twinning plane inclination of $\sim 43^\circ$ for Mg with respect to compression. If we rotate the grain away from this position, the Schmid factor decreases and becomes zero roughly at 49° . If we fix the grain orientation and the loading direction, then the inclination between twin and the loading direction will be different for Mg and Zr due its different c/a values. It is directly reflected in the cut-off angle. The parent grain orientation or twin variant type does not play a role on the m_0 based cut-off angle.

To bring out all these points, we have added a schematic in Fig. 3 along with the corresponding text in the revised manuscript.

"To compare these three geometric factors, a simple bi-crystal setup is considered (shown in Fig. 3(a)). The orientation of the grain with the incoming twin is fixed and its c-axis is normal to the imposed compression. It corresponds to ~ 0.5 Schmid factor for the incoming twin. A total of 221 different orientations are considered for the neighboring grain. All three geometric measures are calculated for all the twin variants of the neighboring grain orientations with respect to the loading direction (for m_0) and/or the incoming twin (for m' and m''), and the maximum geometric measure, which corresponds to the best-oriented twin variant, is selected. The measure m' and m'' range from 0 to 1, and the Schmid factor m_0 range from 0 to 0.5. To have the same range as m' and m'' , we used $2m_0$ in this comparison."

In the second paragraph of page 8, it is stated "The initial activation stresses for these slip families". This "initial activation stress" is typically refereed to the critical resolved shear stress (CRSS), so why not use it here. After all, the abbreviation CRSS is used later in the manuscript, but it has not been defined.

Response: We agree. We have changed it to CRSS.

Does the model incorporate the crystallographically imposed twinning shear?

Response: Yes. The local stress concentrations we predict are induced by the twin shear transformation associated with twinning.

Is the stress that develops in the simulation, shown in figure 5, periodic in the direction perpendicular to the page (i.e. in the x-direction shown in figure 4)? If so, is this the situation in all simulations? If it is, are the simulations limited to case #1 described above, in which the twin planes in both grains share a common intersection with the boundary? If it is not the case, is the stress developed a function of x-direction (i.e. case #2 above). Or does case #1 versus case #2 not matter as the model does not consider discrete dislocation mechanisms and having a twin boundary plane that is not the same as the twinning plane is not an issue?

Response: In the simulations, we have assumed periodic boundary conditions consisting of traction free surfaces in the x and z-directions, and a compression inducing strain in the y-direction. In the model, we consider that the twin boundary plane is the same as the twinning plane. But, from this continuum scale level work, it is not possible to comment on the 3D structure of the ATPs.

Why is TRSS used in some places while in others T-RSS is used in other places.

Response: We agree. We now use only TRSS.

Response to Reviewer – 3

This study focus on a model predicting if an adjacent twin pairs would be formed or not. Combining a large data set of experimental results get by an automated method and a crystal plasticity model, authors compare different HCP materials (mainly Mg and Zr) to show that the transmission criterion depends not only on the missorientation, but also on intrinsic material plastic properties such as the CRSS of slip and twin systems. This study is original and interesting, data and methodology are good, uncertainties explained, conclusion is reliable (except the term 'plastic anisotropy'), Reference ok and the text is clear, however the present reviewer suggest some minor corrections.

Response: We thank the referee for his/her careful and detailed review of our manuscript and for the valuable comments.

1-Abstract: plastic anisotropy : for the reviewer a material with a random orientation is a material plastically isotropic, whereas the meaning of the authors is much different, so I suggest to find a better expression to refer to the plastic properties of grains and single crystals.

Response: What we mean is that the single crystals are plastically anisotropic because the stress to activate $\langle a \rangle$ slip, such as prismatic, is much larger than for $\langle c+a \rangle$ slip. In the abstract, we have replaced **material plastic anisotropy** with **crystal plastic anisotropy**. In addition to that we mention this point in page 8, as,

“Independent of their elastic anisotropy, the plastic deformation of hcp crystals, like Mg and Zr, is anisotropic and carried by multiple slip systems including basal $\langle a \rangle$ slip, prismatic $\langle a \rangle$ slip and pyramidal $\langle c+a \rangle$ slip² with significantly different activation stresses. This leads to anisotropic plastic deformation behavior for single crystals and for textured polycrystals.”

2-introduction p2: 'forming so called twin chains or catalytic twins' : we can also more simply say cross boundary twin or adjacent twin pair [M.R. Barnett, M.D. Nave, A. Ghaderi, Acta Materialia 60 (2012) 1433-1443] or adjoined twin pair as you do later.

Response: When we wrote that sentence, we had in mind that twin chains would involve three or more twins that are connected. We agree that this difference needs to be explained and have added a sentence on page 2. Here we only look at two twins joined at a boundary (ATPs) and acknowledge that we are not studying twin chains

“In this work, we refer to two twins that are connected at the grain boundaries as adjoining twin pairs (ATPs) and those comprised of three or more connected twins as twin chains. The

former is the focus of this study."

3-Figure 1: there is not the direction on the IPF coloring (we guess red is ND), nor the ND and compression direction on the EBSD maps.

Response: Thanks for pointing this out. We have now mentioned the direction of the standard stereographic triangle in Figure 1.

4-p6 section 2.2: 'The rationale for the (Schmid factor) criterion is that if the parent grain with the incoming twin is well suited for twinning under the current loading, any neighboring grain that deviates in orientation from the parent would be less favorable for twinning. Consequently, the further the neighbor deviates from the parent, the less likely it is to twin.' The reviewer disagrees with this explanation: If a parent grain have a SF of 0.5 for 1 twin variant, it could have a neighbor grain (with another orientation) with a SF=0.5 for another twin variant. And this other twin, certainly with a very different plane and shear direction, could join the first twin at the boundary on the EBSD map by chance. In compression this could happen with grains of orientation (0 90 30) and (90 90 30) which are missoriented 90 degree but the SF of the 2 best variant is 0.49.

Response: Reviewer is right in general. However, because {10-12} twins are nearly at 45 degree of the basal plane, and because we base our criterion in selecting the best oriented twin, and because the parent grain twin exhibits the highest possible Schmid factor, this translates into the argument that any neighboring grain that deviates in orientation from the parent would be less favorable for twinning. And we agree that this conclusion applies to {10-12} twins with in-plane compression mode. We have mentioned this in the revised manuscript. For in-plane compression deformation the orientation (0,90,30) [in degrees] puts the c-axis in compression and does not produce a positive Schmid factor (~0.49).

"The rationale for the latest criterion is that if the parent grain with the incoming twin of {10-12} type were well suited for twinning under the current loading, any neighboring grain that deviates in orientation from the parent would be less favorable for twinning. Consequently, the further the neighbor deviates from the parent, the less likely it is to twin."

5- Figure 3: When we read the manuscript and the description, we expect that each point stand for 1 observed ATP, and so are experimental results. But clearly this figure is made with simulations, this is why the points form lines. Also the authors must explain in the text for one pair, how do chose the outgoing twin to compute the Schmid Factor, how do they have a SF m_0 between 1 and 0?

Response: The calculation and the choice of Schmid factor are included in the present version of manuscript in Page 6. The Schmid factor ranges from 0.0 to 0.5. But for the purpose of comparison with the other figures of merit, which range from 0.0 to 1.0, we present two times the Schmid factor $2m_0$. We now clarify this point in the figure caption of Figure 3 and in the corresponding text.

"To compare these three geometric factors, a simple bi-crystal setup is considered (shown in Fig. 3(a)). The orientation of the grain with the incoming twin is fixed and its c-axis is normal to the imposed compression. It corresponds to ~0.5 Schmid factor for the incoming twin. A total of 221 different orientations are considered for the neighboring grain. All three geometric measures are calculated for all the twin variants of the neighboring grain orientations with respect to the loading direction (for m_0) and/or the incoming twin (for m' and m''), and the maximum geometric measure, which corresponds to the best-oriented twin variant, is selected. The measure m' and m'' range from 0 to 1, and the Schmid factor m_0 range from 0 to 0.5. To have the same range as m' and m'' , we used $2m_0$ in this comparison."

6- p7 section 2.2: these geometric indicators (...) do not include materials aspect': for the reviewer's point of view, GB misorientation, grain orientation, twin variant's b and n are all 'material aspects'.

Response: The authors regard GB mis-orientation, grain orientation, twin variant's b and n as geometric parameters and these apply to all studied materials. Parameters like elastic moduli; plastic modes CRSS values and c/a ratio are considered material aspects and they differ between different studied materials. We have mentioned this point explicitly in the manuscript as

"As mentioned earlier, these geometric criteria are only partial indicators for twin transmission because they do not include material aspects like elastic moduli, CRSS values of plastic deformation modes and c/a ratio."

7- p9 section 2.3: as far as we understand, the TRSS is the shear stress along the twin plan and with the twin shear direction in the central grain, but in the 2 neighbor grains, does the shear stress is computed along the same plan which is (usually) not a twin plan or on the best twin variant ?

Response: The TRSS averaged in region Ω is the maximum TRSS among all the six twin variants of the neighboring grain calculated using the local stress and neighboring grain orientation. The TRSS is not calculated with respect to the twin system/variant in the central grain. We now have mentioned this on page 11.

"Using the TRSS as an acceptable measure of the driving forces needed to propagate the twin forward, we calculated the maximum TRSS among all six-twin variants of the neighboring grain."

8-p11 section 2.4: does a ratio $TRSS_{tip}/TRSS_{sx} > 1$ also fit with a higher SF in propagated twin than in the central crystal of orientation 0 0 0 ?

Response: We appreciate the reviewer's question. Actually it does not. The twin in the central grain is the one with the highest Schmid factor (~0.5). The cases where $TRSS_{ratio} > 1$ are due to local stress concentration developed as a consequence of material plastic anisotropy and are not connected with the Schmid factor.

9-p15 conclusion: experimentally for Zr some high misorientation (fig 2b, 80degree) also witness twin transmission, whereas the present calculations predict no transmission. According to Fig3, the geometric factors always expect twin propagation (they are seldom null), so the developed model predict the same tendency as m0 the SF, but with higher cut-off misorientation angle. So the presented simulations (using the 3-crystal) are good to compare different materials but not predict perfectly the creation of adjacent twins.

Response: We agree with reviewer's comment regarding the twin transmission in Zr even at an 80 degree mis-orientation. In the model, we have considered only 3-grains with periodicity in the out-of-plane direction. This assumption and the model set-up may fail to capture some of the twin transmissions. At the same time, the present model correctly predicts the tendency of twin transmission with grain boundary mis-orientation angle and it helps understand the observed differences in twin transmission between HCP Mg and Zr. The observed twin transmission in Zr even at an 80 degree mis-orientation, may be because the twin transmission not only depends on the grain boundary mis-orientation angle. It may depend on the parent and neighboring grain sizes, distribution of crystal orientation in the cluster of grains that surrounds the twinning grains and so on. We are not accounting for these aspects in the present work for twin transmission.

We have added the following paragraph on pages 11/12:

"We note that the EBSD statistical information displayed in Fig. 2 indicates a small number of ATP's in Zr at GBs with misorientation angles larger than this cut-off. We attribute the discrepancy of our model to possible effects of relative differences in the sizes of the parent and neighboring grain and of multi-grain junctions joining two or more neighboring grains with the parent, all of which may induce much larger stress deviations than those generated in the model tri-crystal."

Reviewers' comments:

Reviewer #1 (Remarks to the Author):

The authors have performed an admirable job of responding to all the reviewer comments and addressing key points sensibly within the manuscript. The paper is a great read and a fantastic contribution to the literature.

Reviewer #2 (Remarks to the Author):

As I noted in my previous review, this is a very nice manuscript that further the understanding of correlated twinning across grain boundaries in hexagonal metals. While I previously pointed out a highlight being the conclusion that plastic anisotropy is the critical factor in the twin-to-twin nucleation, I failed to mention the very interesting observation that in some cases the stress concentration that develops at the twin tip can lead to a boost effect in the neighboring grain. This is very interesting.

I am still concerned that in places the manuscript is unclear and difficult to follow. I would have liked the model to be fleshed out a little more. I would also like to see a little more in the way of discussion about the limitations of the model. As I noted in my earlier review, when the full crystallography of twinning transmission across a boundary is taken into account, it complicates the situation as the line defined by the twin plane/boundary intersection is not common for the incoming and outgoing twins if there is any twist component to the boundary. Given their response to my concerns, the authors are clearly comfortable with the realities of this situation and have indicated that they believe the twin junctions extend below the surface. But the paper does not discuss this real possibility and does not indicate that this misalignment might (from a crystallographic sense) influence the process to some degree.

After further considering figure 3 (b) more (line 171 should note figure 3 (b) and not just figure 3), I believe I understand the Schmid factor cut off. Is it true that in actuality at larger misorientations, beyond the cut-off, the Schmid factor actually becomes negative? But because twinning in uni-directional, negative Schmid factors will not lead to twinning (unlike for typical dislocation movement, which will occur in the opposite direction when the Schmid factor is negative), so rather than showing negative Schmid factors, the authors have are just considered negative values to be zero? If this is the case, beyond the cut-off, would the resistance to twin transfer would increase? If I am interpreting this correctly, perhaps a footnote or a statement in the caption indicating that negative Schmid factors are considered zero would be appropriate.

In figure 2 b, there is still some observations of twinning transfer beyond the “cut-off”. What other factors might be causing the observation of some twin transfer at these high misorientation angles? Could this be discussed?

Reviewer #3 (Remarks to the Author):

All our questions have been answered satisfactorily and our remarks have been taken into account, so we recommend the article to be published.

Response to Reviewer - 1

The authors have performed an admirable job of responding to all the reviewer comments and addressing key points sensibly within the manuscript. The paper is a great read and a fantastic contribution to the literature.

Response: We thank the referee for his/her comment.

Response to Reviewer - 2

As I noted in my previous review, this is a very nice manuscript that further the understanding of correlated twinning across grain boundaries in hexagonal metals. While I previously pointed out a highlight being the conclusion that plastic anisotropy is the critical factor in the twin-to-twin nucleation, I failed to mention the very interesting observation that in some cases the stress concentration that develops at the twin tip can lead to a boost effect in the neighboring grain. This is very interesting.

Response: We thank the referee for his/her kind words and for the careful and detailed review of our manuscript. We found the comments valuable toward improving the manuscript. Following his/her suggestions we added in the appropriate places the requested clarifications and discussions.

I am still concerned that in places the manuscript is unclear and difficult to follow. I would have liked the model to be fleshed out a little more. I would also like to see a little more in the way of discussion about the limitations of the model.

As I noted in my earlier review, when the full crystallography of twinning transmission across a boundary is taken into account, it complicates the situation as the line defined by the twin plane/boundary intersection is not common for the incoming and outgoing twins if there is any twist component to the boundary. Given their response to my concerns, the authors are clearly comfortable with the realities of this situation and have indicated that they believe the twin junctions extend below the surface. But the paper does not discuss this real possibility and does not indicate that this misalignment might (from a crystallographic sense) influence the process to some degree.

Response: We appreciate the reviewer's comments, both regarding inclusion of discussion on model limitations and on the likelihood of subsurface intersections. Regarding the first one, due to space limitations in the manuscript, authors discuss the FFT model formulation and its limitations in the added supplementary.

With respect to reviewer's concern regarding the crystallography of twinning transmission across a boundary, we acknowledge its validity. We do not have definitive evidence showing that the Adjoining Twin Pairs (ATP's) observed in the EBSD scans of Fig 1 represent 3D features that continue below the surface of the sample. It is our intention, in the near future, to perform EBSD combined with serial sectioning in order to elucidate this issue (as well as the 3D structure of twin-twin junctions inside the grain). For the

purpose of this review we would like to propose a simple mechanism in support of our statement, and a qualitative argument against the non-continuation of the joint twin trace below the surface.

Figure 1 below depicts two crystals with a slight tilt of the $\{10\bar{1}2\}$ plane, which is the case for most ATPs present in the EBSDs of Fig 1 in the manuscript. The way we envision transmission across the grain boundary is as follows:

1) The propagating twin (left crystal) impinges on the GB and generates twin nuclei along the trace (we have characterized such nucleation mechanisms in [Wang, Beyerlein, Tomé, “An atomic and probabilistic perspective on twin nucleation in Mg”, *Scripta Materialia* 63 (2010) 741-746]);

2) Next, $\{10\bar{1}2\}$ twins are emitted separately from the various nucleation sites, propagate forward on $\{10\bar{1}2\}$ planes, and coalesce into a bigger twin domain. The interface of the latter is formed mainly by $\{10\bar{1}2\}$ coherent twin planes plus semi-coherent serrations made of prism-prism planes (we have characterized the latter in: Liu, Li, Shao, Gong, Wang, McCabe, Jiang, Tomé, “Characterizing the boundary lateral to the shear direction of deformation twins in magnesium”, *Nature Communications* 7,11577 (2016) 1-6);

3) Such a ‘corrugated twin’ would have the same trace as the one on the left, and serrations (nano-metric) would not be observable with EBSD resolution.

Figure 1

Figure 2 shows the traces in the GB of parallel $\{10\bar{1}2\}$ twin lamellae in the left and right crystal, assumed to have been generated independently (as opposed to being transmitted across the boundary). The question is: how likely it is, in such situation, that a section of the deformed sample will cut across one of such intersections? In this example, a total of 11 trace intersections can be identified. If the GB is sectioned (for observation) by planes perpendicular to it (red dash lines), the chance of intersecting one of the 11 trace junctions is low (between 0 and 2, depending on the section chosen). While our argument is

qualitative from the stereological and probabilistic analysis, the plausible conclusion is that hitting trace intersections has low probability and, in addition, missing them closely would reveal non-fully superposed (off-set) twin junctions, such as the one shown in Figure 2, which is not the case in our EBSDs.

Figure 2

Unfortunately, space limitations in the manuscript do not allow us to include the lengthy discussion above. Instead, we have addressed the issue as follows (in page 4):

“The continuation of the ATPs below the EBSD scan surface cannot be guaranteed, and performing serial sectioning would be an appropriate way to confirm it. However, if adjoining twin pairs were 1D features, the chance that a section of the deformed sample would cut across one such trace intersection would be exceedingly small. On the other hand, previous experimental observation [Beyerlein et al., 2010] shows that the frequency of observing ATPs in a particular surface is quite high”

After further considering figure 3 (b) more (line 171 should note figure 3 (b) and not just figure 3), I believe I understand the Schmid factor cut off. Is it true that in actuality at larger misorientations, beyond the cut-off, the Schmid factor actually becomes negative? But because twinning in uni-directional, negative Schmid factors will not lead to twinning (unlike for typical dislocation movement, which will occur in the opposite direction when the Schmid factor is negative), so rather than showing negative Schmid factors, the authors have are just considered negative values to be zero? If this is the case, beyond the cut-off, would the resistance to twin transfer would increase? If I am interpreting this correctly, perhaps a footnote or a statement in the caption indicating that negative Schmid factors are considered zero would be appropriate.

Response: Figure 3(b) now replaces Figure 3 in line 171. Many thanks for your valuable observation/suggestion regarding negative Schmid factor. The Schmid factor for larger misorientation cases is negative and we report those to be zero, to consider only the forward direction. We have mentioned this point in the revised Figure 3 caption as:

“Note that the Schmid factor for the larger misorientation cases ($> \sim 50^\circ$) can be negative, in which case we assign a zero value to it, to consider only the forward twin shear direction.”

In figure 2 b, there is still some observations of twinning transfer beyond the “cut-off”. What other factors might be causing the observation of some twin transfer at these high misorientation angles? Could this be discussed?

Response: The observed twin transmission beyond the cut-off angle shown in Fig. 2(b) can be a consequence of uncertainty in the statistical analysis or, more likely, the manifestation of local stress conditions associated with clusters of grains, which add to the complex stochastic nature of twinning. Note that such situation is observed (Figure 2 of the manuscript) in small-grained Zr and not in large-grained Mg. As a consequence, we can probably rule out uncertainties in the automated analysis. In what concerns inhomogeneity: clusters of small grains (Zr) are likely to present larger stress deviations with respect to the macroscopic stress than clusters of large grains (Mg). As a consequence, twins may be activated that have negative Schmid factor (when the latter is defined with respect to the macroscopic loading). This point is now included in the revised Figure 2 caption as:

“Non-zero crossed boundary fraction beyond 60° for Zr can be a consequence of local stress deviations associated with clusters of small-sized grains where twin variants with negative ‘macroscopic’ Schmid factor are activated”

We also mention this point in connection with model in the revised manuscript page 11, and it read as:

“We note that the EBSD statistical information displayed in Fig. 2 indicates a small number of ATP’s in Zr at GBs with misorientation angles larger than this cut-off. We attribute the discrepancy to possible large stress deviations associated with clusters of small-sized Zr grains, where twin variants with negative ‘macroscopic’ Schmid factor are activated. Such deviations are not generated in the model tri-crystal”

Response to Reviewer – 3

All our questions have been answered satisfactorily and our remarks have been taken into account, so we recommend the article to be published.

Response: We thank the referee for his/her comments.

Reviewers' Comments:

Reviewer #2 (Remarks to the Author):

My primary concerns have been addressed by the authors.

Response to Reviewer - 1

None

Response to Reviewer – 2

My primary concerns have been addressed by the authors.

Response: We thank the referee for his/her comments.

Response to Reviewer – 3

None